# Novel Sustained-Release Drug Delivery System for Dry Eye Therapy by Rebamipide Nanoparticles

**DOI:** 10.3390/pharmaceutics12020155

**Published:** 2020-02-14

**Authors:** Noriaki Nagai, Miyu Ishii, Ryotaro Seiriki, Fumihiko Ogata, Hiroko Otake, Yosuke Nakazawa, Norio Okamoto, Kazutaka Kanai, Naohito Kawasaki

**Affiliations:** 1Faculty of Pharmacy, Kindai University, 3-4-1 Kowakae, Higashi-Osaka, Osaka 577-8502, Japan; 1833420012r@kindai.ac.jp (M.I.); 1611610157u@kindai.ac.jp (R.S.); ogata@phar.kindai.ac.jp (F.O.); hotake@phar.kindai.ac.jp (H.O.); kawasaki@phar.kindai.ac.jp (N.K.); 2Faculty of Pharmacy, Keio University, 1-5-30 Shibakoen, Minato-ku, Tokyo 105-8512, Japan; nakazawa-ys@pha.keio.ac.jp; 3Okamoto Eye Clinic, 5-11-12-312 Izumicho, Suita, Osaka 564-0041, Japan; eyedoctor9@msn.com; 4Department of Small Animal Internal Medicine, School of Veterinary Medicine, University of Kitasato, Towada, Aomori 034-8628, Japan; kanai@vmas.kitasato-u.ac.jp

**Keywords:** rebamipide, sustained delivery system, dry eye, eyelid, mucin

## Abstract

The commercially available rebamipide ophthalmic suspension (CA-REB) was approved for clinical use in patients with dry eye; however, the residence time on the ocular surface for the traditional formulations is short, since the drug is removed from the ocular surface through the nasolacrimal duct. In this study, we designed a novel sustained-release drug delivery system (DDS) for dry eye therapy by rebamipide nanoparticles. The rebamipide solid nanoparticle-based ophthalmic formulation (REB-NPs) was prepared by a bead mill using additives (2-hydroxypropyl-β-cyclodextrin and methylcellulose) and a gel base (carbopol). The rebamipide particles formed are ellipsoid, with a particle size in the range of 40–200 nm. The rebamipide in the REB-NPs applied to eyelids was delivered into the lacrimal fluid through the meibomian glands, and sustained drug release was observed in comparison with CA-REB. Moreover, the REB-NPs increased the mucin levels in the lacrimal fluid and healed tear film breakup levels in an *N*-acetylcysteine-treated rabbit model. The information about this novel DDS route and creation of a nano-formulation can be used to design further studies aimed at therapy for dry eye.

## 1. Introduction

Dry eye is a multifactorial disease of the tears and the ocular surface that results in symptoms of discomfort, visual disturbance, and tear film instability with potential damage to the ocular surface. It is accompanied by increased osmolarity of the tear film and inflammation of the ocular surface [1]. In addition, the negative effects that dry eye have on visual function, quality of life, and economic burden are well recognized [2,3]. In many patients, the condition is chronic and requires long-term treatment, and potentially more effective ophthalmic pharmacological drugs targeting various distinct pathophysiological pathways of dry eye have been investigated. In Japan, the formulation to enhance the aqueous humor and mucin secretion are mainly used in the therapy of dry eye.

Rebamipide has followed a unique course in drug discovery and has long been used as a treatment for gastric ulcers. The logP and pK of rebamioide are 2.9 and 3.3, respectively, and the Biopharmaceutical Classification System (BCS) lists rebamipide as a class IV drug. In recent years, its mucosal-protective effect has also been applied to protection of the keratoconjunctival epithelium [4,5] after the development of ophthalmic rebamipide products for the treatment of dry eye [4,5,6,7]. With regard to the pharmacological mechanisms of rebamipide in dry eye, the majority of studies have focused on mucin production. The commercially available rebamipide ophthalmic suspension (CA-REB, Mucosta Ophthalmic Suspension UD 2%, Otsuka Pharmaceutical, Co., Ltd., Tokyo, Japan) was approved for the treatment of dry eye at the end of 2011 and was launched in Japan in 2012. In the clinical trial, the instillation of rebamipide was performed four times/day for the patient with dry eye, and in clinical studies, rebamipide has been demonstrated to be effective in improving the symptoms and signs of dry eye [6,7,8]. The use of rebamipide has been extended to dry eye treatment due to the discovery of its ocular surface mucin-increasing action. Previous studies showed that topical rebamipide may increase the number of goblet cells and promote the secretion of mucin-like substances in the bulbar conjunctiva and lacrimal caruncle of humans [9,10], and rebamipide has been found to improve both vital staining and tear film breakup time.

The ophthalmic application of drugs is the primary route of administration for the treatment of various eye diseases and is well-accepted by patients. However, in traditional formulations, only small amounts of the administered drug reach their target due to dilution caused by lacrimation and evacuation through the nasolacrimal duct [11]. Consequently, frequent instillation is needed to obtain a sufficient therapeutic effect. Eye ointment formulations are also used in the clinic. However, there are problems of convenience with the application of an eye ointment. Therefore, it is very important to design a sustained drug delivery system (DDS) in the ophthalmic field.

Solid drug nanoparticles come with the added benefits of possible cellular targeting and improvement in cellular uptake and have been used widely as nanotechnology-based delivery systems. We previously designed solid nanoparticles created by a breakdown method (bead mill), and reported on their low toxicity and high transdermal penetration via endocytosis when used in nano-formulations [12,13,14,15,16]. It is expected that the application of solid nanoparticles to the eyelid may be a possible route to sustained drug supplementation to the ocular surface and provide a novel strategy for ophthalmic DDS. In this study, we attempted to design a rebamipide nano-DDS through the eyelid and evaluate its usefulness for dry eye treatment.

## 2. Materials and Methods

### 2.1. Animals

Adult rabbits (male, weight 2.71 ± 0.43 kg, *n* = 26) were used in experiments performed according to the guidelines for The Association for Research in Vision and Ophthalmology (ARVO) and the protocol approved by the Pharmacy Committee Guidelines for the Care and Use of Laboratory Animals in Kindai University (KAPS-25-003, 1 April 2013). The rabbit model of dry eye was obtained by the instillation of 10% *N*-acetylcysteine (dry eye model), and 1.5% REB formulations (0.3 g) were applied to the shaved eyelid skin in single or repetitive applications at 14:00. For the repetitive applications, 1.5% rebamipide formulations (0.3 g) were applied once a day (14:00) for six days, and the measurements of lacrimal fluid volume, mucin levels, tear film breakup time (TBUT), ocular surface, and tea film breakup levels were started at 18:00.

### 2.2. Preparation of Rebamipide Solid Nanoparticle-Based Ophthalmic Formulations (REB-NPs)

Ophthalmic dispersions containing rebamipide nanoparticles were prepared following the previous reports [12,13,14,15,16]. Briefly, rebamipide powder (particle size 741 ± 12.7 nm) purchased from Wako Pure Chemical Industries, Ltd. (Osaka, Japan) was mixed with 2-hydroxypropyl-β-cyclodextrin (HPβCD, Nihon Shokuhin Kako Co., Ltd., Tokyo, Japan) and type SM-4 methylcellulose (MC, Shin-Etsu Chemical Co., Ltd., Tokyo, Japan) in distilled water, and the dispersions were milled at 5500 rpm for 1 min × 30 times using 0.1 mm zirconia beads and a Micro Smash MS-100R (TOMY SEIKO Co. Ltd., Tokyo, Japan). The milled mixtures were gelled with carboxypolymethylene (Carbopol^®^ 934, carbopol, Serva, Heidelberg, Germany) and used as REB-NPs. The preparation of the rebamipide powder (solid microparticle)-based ophthalmic formulation was performed according to same protocol without the bead mill treatment (REB-MPs). The compositions of REB-MPs and REB-NPs were as follows: 1.5% rebamipide, 5% HPβCD, 0.5% MC, carbopol, in distilled water.

### 2.3. Measurement of Rebamipide Levels

The rebamipide in samples was extracted with *N*,*N*-dimethylformamide on ice and measured on an HPLC LC-20AT system (Shimadzu Corp. Kyoto, Japan). The HPLC conditions were as follows: wavelength, 287 nm; temperature, 35 °C; internal standard, 1 μg/mL methyl p-hydroxybenzoate; mobile phase, 50 mM phosphate buffer/acetonitrile (75/25, *v*/*v*); flow rate, 0.25 mL/min; column, 2.1 × 50 mm Inertsil^®^ ODS-3 column (GL Science Co., Inc., Tokyo, Japan). The detection limit of HPLC was 70.4 ng/mL, and the R value was 0.9992 in the calibration curve.

### 2.4. Evaluation of Rebamipide Particles in REB Formulations

A nanoparticle size analyzer laser diffraction SALD-7100 (Shimadzu Corp.) with the refractive index set to 1.60–0.10i was used to measure the size distribution of rebamipide particles in REB-MPs and REB-NPs. The size distribution and number of nanoparticles in REB-NPs were analyzed by a dynamic light scattering NANOSIGHT LM10 (QuantumDesign Japan, Tokyo, Japan). The measurement time was as 60 s, and wavelength and viscosity of the suspension were set to 405 nm (blue) and 1.27 mPa·s, respectively. A scanning probe microscope SPM-9700 (Shimadzu Corp.) was used to obtain an atomic force microscopic (AFM) image in this study.

### 2.5. Dispersity and Stability in REB Formulations

REB-MPs and REB-NPs, 0.3 g each, were divided into 10 parts, and the rebamipide content in each part was measured to investigate dispersity. In addition, the REB-MPs and REB-NPs preparations were kept at 25 °C for one month under dark conditions to measure stability. The size distribution and concentration for demonstrating dispersity and stability were determined by the SALD-7100, NANOSIGHT, and HPLC methods described above.

### 2.6. Rebamipide Release from REB Formulations

A membrane filter and Franz diffusion cell were used to evaluate the release of rebamipide from REB formulations [13]. The reservation chamber of the diffusion cell was filled with 12.2 mL of 10 mM phosphate, and a 25 nm- or 450 nm-pore size MF™-MEMBRANE FILTER (Merck Millipore, Tokyo, Japan) was set into Franz diffusion cell to which 0.3 g of the 1.5% REB formulations was applied gently. The area under the rebamipide concentration-time curve (*AUC*_Release_) was analyzed by the trapezoidal rule for the data for 0–24 h, and the size distribution of nanoparticles and concentration in the reservoir chamber were determined by the NANOSIGHT and HPLC methods described above.

### 2.7. Rebamipide Levels in Lacrimal Fluid and Meibum of Rabbits Applied with REB Formulations

REB-MPs or REB-NPs formulations (1.5%; 0.3 g) were applied to the shaved eyelid of rabbits, and the meibum and lacrimal fluid without meibum were collected with Schirmer tear test strips. The lacrimal fluid without meibum was harvested as follows: space was made between the eyelid and the ocular surface of a rabbit, and the Schirmer tear test strips were attached to the eyelid side (conjunctival sac). The Schirmer tear test strips containing samples were homogenized in *N*,*N*-dimethylformamide, and the rebamipide was extracted. The rebamipide concentrations were determined by HPLC as described above, and the *AUC* for rebamipide levels in lacrimal fluid (*AUC*_LF_) were analyzed by the trapezoidal rule up to 180 min. 

### 2.8. Monitoring the Ocular Surface of Rabbits Applied with REB Formulations

Schirmer tear test strips were used to measure the volume of lacrimal fluid in rabbits applied with REB-MPs and REB-NPs. The TBUT and changes in the ocular surface were measured 6 h after the application of REB formulation (18:00). A rabbit treated with a fluorescein strip was allowed to blink several times to distribute the fluorescein. The time from opening of the eyes to the appearance of the first dry spot in the central cornea was analyzed, and the time was presented as TBUT. The measurement was performed three times, and the mean was used as the value. The changes in tear film after winkling were monitored, and tear film breakup levels were evaluated by dry eye monitor DR-1 (KOWA Co., LTD., Aichi, Japan).

### 2.9. Mucin Levels in Rabbits Applied with REB Formulations

The lacrimal fluid was collected by Schirmer tear test strips and homogenized in *N*,*N*-dimethylformamide. The mucin in the supernatants was measured using a tear mucin assay ELISA kit (Cosmo Bio Co., Ltd., Tokyo, Japan) according to the manufacturer’s instructions. A fluorescence microplate reader was used to measure the mucin levels (Absorption/Emission = 336 nm/383 nm), which are expressed as the ratios to the mucin levels at the start of the experiment (normal rabbit, 0.69 ± 0.06 mg/mL, *n* = 24; rabbit model with dry eye, 0.42 ± 0.03 mg/mL, *n* = 27).

### 2.10. Statistical Analysis

Statistical data from the SALD-7100 are expressed as the mean ± S.D., and other data are expressed as the mean ± S.E. Differences between mean values were analyzed with ANOVA followed by the Student’s *t*-test and Dunnett’s multiple comparisons. P-values less than 0.05 were considered significant.

## 3. Results

### 3.1. Design of a Rebamipide Solid Nanoparticle-Based Ophthalmic Formulation

Our previous study showed that bead mill treatment with MC allows a decreased particle size to the nano level, and that the addition of HPβCD prevents aggregation of the nanoparticles [12,13,14,15,16]. In addition, we also reported that carbopol is suitable as a base for dermal formulations [13]. Taken together, we attempted to prepare rebamipide nanoparticles based on our previous studies using additives (HPβCD and MC) and a gel base (carbopol). Rebamipide particles (approximately 100 nm–25 μm) were crushed by mill treatment. The milled rebamipide nanoparticles had a size of approximately 40–200 nm in the carbopol gel (Figure 1A,B, Appendix A) and were ellipsoid in form (Figure 1C). The rebamipide solubility increased by approximately 3.4-fold with the decrease in particle size, although the solubility remained low, and 99.92% still existed as solid particles (Figure 1D). On the other hand, the solubility of REB-MPs and REB-NPs without HPβCD were 0.003 fM, 0.011 fM, respectively, and it was suggested that the both of nano crystallization and enhanced inclusion complexes with HPβCD were related the increase of drug solubility in the REB-NPs.

Figure 2 shows the changes in REB-NPs one month after preparation. No differences were observed in the size or form of the rebamipide nanoparticles in REB-NPs (Figure 2A–C), and the ratio of solid to solution in REB-NPs was similar one month after preparation as it had been immediately (day 0) after preparation (Figure 2D). In addition, the rebamipide particles were distributed more evenly in the REB-NPs than in REB-MPs (S.D., REB-MPs 0.0571%, REB-NPs, 0.0034%, *n* = 6). No degradation was observed one month after preparation (rebamipide content of 1 month/0 month = 99.9%).

Figure 3 shows the release of rebamipide particles from the REB formulation in the in vitro study. Dissolved rebamipide was detected in both the REB-MPs and REB-NPs formulations, but no nanoparticles were detectable in the reservoir chamber when it was separated from the sample chamber by a 25 nm membrane filter (Figure 3A and Appendix A). On the other hand, when the chambers were separated by a 450 nm membrane filter, rebamipide nanoparticles were released from REB-NPs (Figure 3B, Appendix A), and both dissolved rabamipide and solid nanoparticles were detected after the application of REB-NPs. The reservoir chamber was found to contain 3.36×10^11^ particles (Figure 3C,D), and the plateau showed that all rebamipide in the REB-NPs sifted to reservoir side. Otherwise, the drug release from REB-NPs was slow in comparison with formulation containing dissolved rebamipide (Appendix A).

### 3.2. Drug Delivery of REB-NPs through the Eyelid 

It is important to investigate the drug delivery route for the novel rebamipide solid nanoparticle-based ophthalmic formulations. Figure 4 shows the trans-eyelid penetration of rebamipide in rabbits to which REB-NPs was applied. Although no residual rebamipide can be detected 60 min after the instillation of commercially available eye drops, the rebamipide shifted from the eyelid to the lacrimal fluid of rabbits treated with REB-NPs (Figure 4A) with an *AUC*_LF_ that was 28.8-fold higher than that in rabbits to which REB-MPs was applied (Figure 4B). On the other hand, little penetrated rebamipide was present in the lacrimal fluid without meibum, and almost all of the penetrated rebamipide was detected in the meibum (Figure 4C). In addition, no solid nanoparticles were detected in meibum or lacrimal fluid with meibum by the dynamic light scattering measurement. These results suggested that the rebamipide was dissolved in the meibum, and meibum may show a high binding affinity to hydrophobic drugs.

### 3.3. Therapeutic Potential of the REB-NPs for Dry Eye

Next, we demonstrated the usefulness of REB-NPs as therapy for dry eye. Figure 5A,B show the lacrimal fluid volume (Figure 5A) and mucin levels (Figure 5B) after the application of REB-NPs to the eyelid (Appendix A). The lacrimal fluid volume and mucin levels were significantly increased by the application of REB-NPs, and at 6 h after application, the lacrimal fluid volume and mucin levels were 2.0-fold and 2.1-fold greater in comparison with the non-treatment group, respectively. Moreover, the TBUT was increased 1.3-fold by the application of REB-NPs as compared with the non-treatment group (Figure 5C and Appendix A). Figure 5D shows the condition of the ocular surface under a Noncontact Specular Microscope (DR-1). Although the Grade level based on the tear oil zone was Grade 4 30 min after eyelid opening in the non-treatment group, the Grade level in rabbits treated with REB-NPs remained Grade 2 at the corresponding time (Figure 5D). Figure 6 shows the changes in the lacrimal fluid volume, mucin level, and tear film breakup levels in the *N*-acetylcysteine-treated dry eye model rabbits treated with or without REB-NPs. A decrease in mucin level and strong tear film breakup levels were caused by treatment with acetylcysteine, and this damage to the ocular surface still persisted 6 days later. On the other hand, the repetitive application of REB-NPs enhanced the repair rate of the ocular surface damage: mucin levels normalized, and the lacrimal fluid volume increased (Figure 6A,B). Moreover, the tear film breakup levels were decreased 6 days after treatment (Figure 6D).

## 4. Discussion

CA-REB was approved for clinical use in dry eye patients [17] even though the residence time is short since the drug is diluted by lacrimation after instillation of the traditional ophthalmic formulation and removed from the ocular surface through the nasolacrimal duct [11]. Therefore, it is expected that the design of a novel ophthalmic DDS will make possible sustained drug supplementation onto the ocular surface. In this study, we developed a novel rebamipide solid nanoparticle-based ophthalmic formulation (REB-NPs), and clarified its penetration route to the ocular surface: rebamipide applied to the eyelid shifts to the ocular surface through the meibomian glands. Moreover, we found that drug supplementation from REB-NPs is sustained and that REB-NPs appears to provide a useful therapy for dry eye (Figure 7).

There are many methods for preparing solid drug nanoparticles by break down and build up, and we also previously reported a break down-based method using special additives [12,13,14,15,16]. The selection of additives is important for preparing REB-NPs, and MC and HPβCD lead to an enhancement in the crushing force and dispersion stability, respectively [12,13,14,15,16]. According to these findings, MC and HPβCD were used to prepare rebamipide nanoparticles, and the rebamipide nanoparticles were gelled by carbopol. The particle size of the rebamipide solid nanoparticles in the REB-NPs formulation was in the range of 40–200 nm (Figure 1B,C), and almost all of it was present in the solid form (Figure 1D). In addition, the rebamipide solid in the REB-NPs remained nano-size, with no differences observed in shape, solubility, dispersity, or content for 1 month after preparation (Figure 2). We demonstrated the release of rebamipide particles from the rebamipide formulation using membrane filters and a Franz diffusion cell (Figure 3). There was more drug released from the REB-NPs than from the REB-MPs (Figure 3A,B), and solid nanoparticles were detected in the reservoir chamber (Figure 3C,D). These results show that REB-NPs is stable, and that rebamipide solid is released as nanoparticles from the gel base.

Next, the drug delivery route was investigated in rabbits to which REB-NPs was applied to the eyelid. The rebamipide in REB-NPs penetrated to the lacrimal fluid side, and sustained release was observed in comparison with traditional rebamipide eye drops. No residual rebamipide can be detected 60 min after the instillation of commercially available eye drops, but when the REB-NPs was applied it the eyelid, rebamipide was still detected for more than 180 min after application (Figure 4A,B). Moreover, the high rebamipide levels were detected in the meibum (lipid), and the penetrated rebamipide dissolved in the meibum and lacrimal fluid without meibum. The meibomian glands are large sebaceous glands located in the eyelid that secrete meibum into the tear film in order to prevent excessive evaporation of lacrimal fluid [18]. Meibum shows a high binding affinity to hydrophobic drugs, such as rebamipide, since meibum is lipid, and we showed that the rebamipide was dissolved in the meibum in this study. From these results, it is hypothesized that the rebamipide in the REB-NPs penetrates into the eyelid, and shifts to the meibomian glands with high affinity. After that, the dissolved rebamipide in the meibum is delivered to the lacrimal fluid. On the other hand, we previously reported that some forms of endocytosis, such as caveolae-dependent endocytosis, clathrin-dependent endocytosis and micropinocytosis, are related to the transdermal penetration mechanism of solid nanoparticles in the epidermis layer of skin [13]. In addition, rabbits have a lot of hair follicles that provide openings in the skin. Therefore, nanoparticle permeation in that case may be much higher than in the human skin. Further studies are needed to evaluate the delivery mechanism of REB-NPs in the human eyelid.

It is important to demonstrate the therapeutic effect of REB-NPs for dry eye. Dry eye is caused by decreases in mucin secretion and lacrimal fluid volume on the ocular surface. Mucin has many important roles on the ocular surface (i.e., lacrimal fluid maintenance, lubrication of the ocular surface to facilitate smooth blinking, formation of a smooth spherical surface for good vision, provision of a barrier for the ocular surface, and the trapping and removal of pathogens and debris) [17,18,19]. Thus, dry eye is defined as a multifactorial disease of the tears and ocular surface. I was previously demonstrated that the topical administration of rebamipide increases mucin levels in the tear film and improves the condition of the ocular surface in dry eye [6]. Therefore, the volume of lacrimal fluid and mucin levels after the application of REB-NPs were measured in this study. REB-NPs enhanced both the lacrimal fluid volume and mucin levels on the ocular surface (Figure 5A,B), and the TBUT was also increased (Figure 5C). These results support previous reports about rebamipide and the drug behavior data (Figure 4) in this study.

It is necessary to investigate the therapeutic effect in a dry eye model. *N*-acetylcysteine is generally recognized to cause a shift to low-molecular-weight mucin molecules by breaking mucoprotein disulfide bonds. The instillation of *N*-acetylcysteine leads to histologic reductions in the mucin layer covering the cornea and conjunctiva, elimination of microvilli, and desquamation of corneal and conjunctival epithelial cells [20], and the thickness of the tear fluid layer is reduced [21]. In addition, it has been reported that the instillation of rebamipide increases the number of conjunctival goblet cells in normal rabbits [22], and the mucin-like substance content on the ocular surface in the *N*-acetylcysteine-treated rabbit model [4]. These findings suggest that the *N*-acetylcysteine-treated rabbit model is suitable for evaluating the therapeutic effect of rebamipide. The repetitive application of REB-NPs attenuated the decrease in mucin levels and the lacrimal fluid volume was increased in the eyes of the *N*-acetylcysteine model rabbits (Figure 6A,B). Moreover, the ocular surface normalized in the model rabbits (Figure 6C), and the tear film breakup levels were enhanced after six days of repetitive application of REB-NPs (Figure 6D). These results show that the REB-NPs are useful as therapy for dry eye. Further studies are needed to develop a rebamipide nano-delivery system through the eyelid, and it is important to clarify the mechanism of the transeyelid penetration of REB-NPs. Therefore, we will investigate the relationships of endocytosis pathways with the transeyelid penetration of REB-NPs using endocytosis-specific inhibitors such as nystatin, dynasore, rottlerin and cytochalasin D.

## 5. Conclusions

We designed a rebamipide solid nanoparticle-based ophthalmic formulation (REB-NPs) and showed that it provides sustained rebamipide supplementation to the ocular surface through the meibomian glands. In addition, REB-NPs provided a highly therapeutic treatment for dry eye, probably caused by an enhancement in mucin levels. Significant information about the novel route of transport through the meibomian glands, and the highly effective therapy for dry eye can be used to design further studies aimed at discovering new therapies for dry eye.

## Figures and Tables

**Figure 1 pharmaceutics-12-00155-f001:**
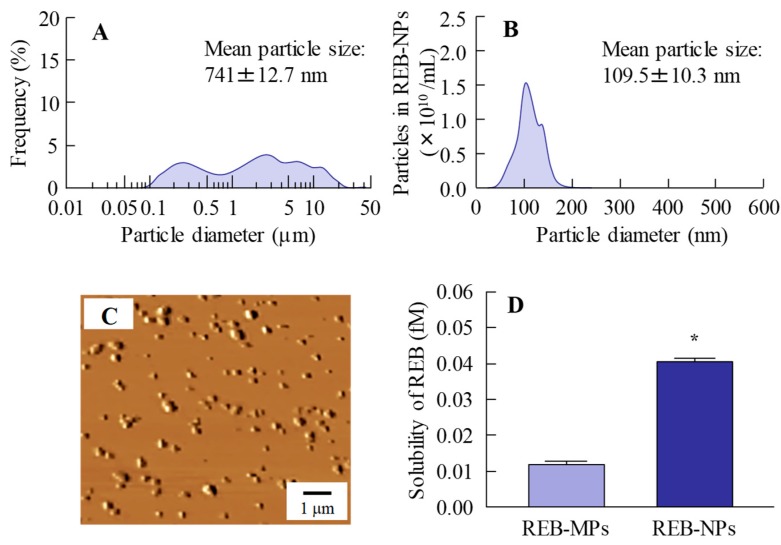
Characterization of the particle size, shape and solubility of rebamipide solid microparticles (REB-MPs) and nanoparticle (REB-NPs)-based ophthalmic formulation. (**A**) REB-MPs size by laser diffraction measurement. (**B**) REB-NPs size by dynamic light scattering measurement. (**C**) AFM image of REB-NPs. (**D**) Drug solubility in REB-MPs and REB-NPs. *n* = 6. **P* < 0.05, vs. REB-MPs. 99.92% of the rebamipide existed in the solid form in REB-NPs, and the mean particle size was 109.5 nm.

**Figure 2 pharmaceutics-12-00155-f002:**
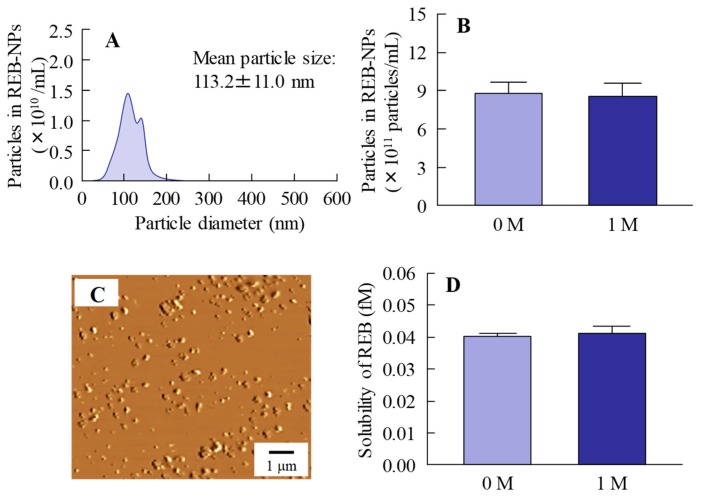
Stability of rebamipide solid in REB-NPs one month after preparation. (**A**) REB-NPs size by dynamic light scattering measurement. (**B**) Number of rebamipide nanoparticles in REB-NPs. (**C**) AFM image of REB-NPs. (**D**) Drug solubility in REB-MPs and REB-NPs. *n* = 6. The rebamipide solid in REB-NPs remained in the nano-size range, and no difference was observed in either the shape or solubility after one month.

**Figure 3 pharmaceutics-12-00155-f003:**
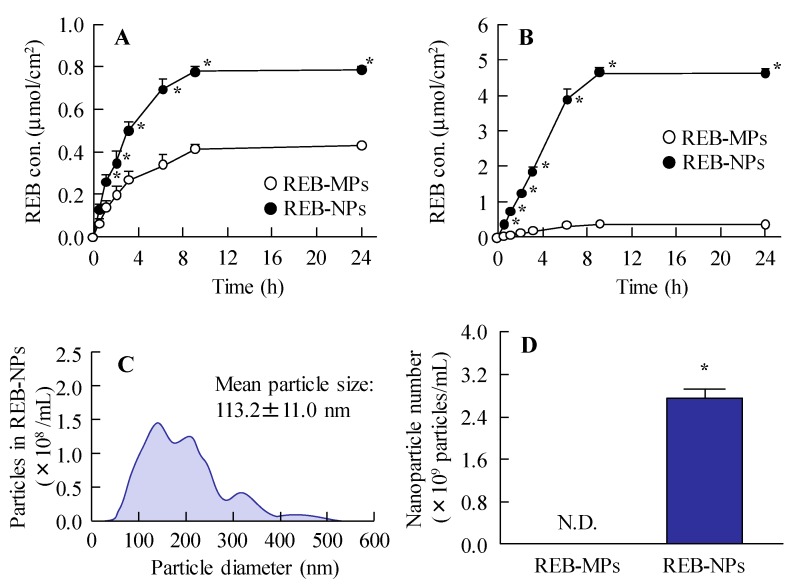
Rebamipide release from REB-MPs and REB-NPs through 25 nm and 450 nm pore membranes. Drug release from REB-MPs and REB-NPs through (**A**) 25 nm and (**B**) 450 nm pore membranes. (**C**) Particle size and (**D**) number of rebamipide nanoparticles that passed through the 450 nm pore membrane 24 h after the application of REB-NPs. Data show the size distribution and number of nanoparticles in the reservoir chamber. *n* = 5–6. N.D., not detectable. **P* < 0.05, vs. REB-MPs. The rebamipide solid was released as nanoparticles from REB-NPs.

**Figure 4 pharmaceutics-12-00155-f004:**
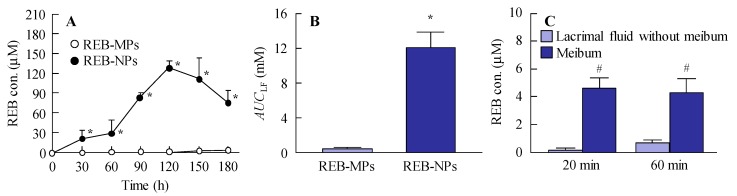
Changes in rebamipide levels in the lacrimal fluid and meibum of rabbits receiving a single treatment of REB-NPs or REB-MPs. (**A**) Rebamipide profile and (**B**) *AUC*_LF_ in the lacrimal fluid after the application of REB-MPs or REB-NPs. (**C**) Rebamipide levels in the meibum and lacrimal fluid without meibum after the application of REB-MPs or REB-NPs. Here, 20 min and 60 min after the application of REB-NPs, the lacrimal fluid without meibum was collected from the eyelid side using Schirmer tear test strips. *n* = 5–7. **P* < 0.05, vs. REB-MPs for each group. ^#^*P* < 0.05, vs. lacrimal fluid without meibum for each group. The rebamipide in the REB-NPs penetrated the eyelid, and was delivered to the lacrimal fluid through the meibomian glands.

**Figure 5 pharmaceutics-12-00155-f005:**
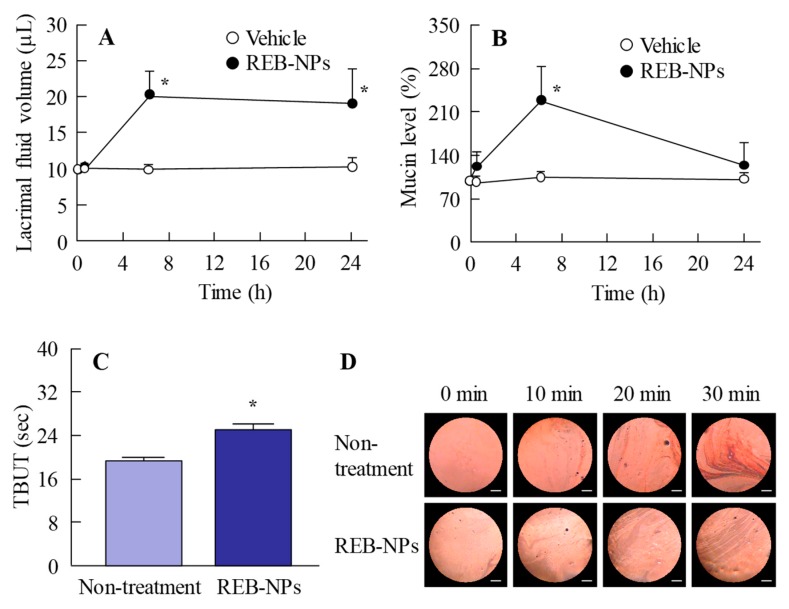
Effect of a single application of REB-NPs on lacrimal fluid volume, mucin levels, and TBUT in rabbits. (**A**) Changes in lacrimal fluid volume and (**B**) mucin levels after the application of REB-MPs or REB-NPs. (**C**) Changes in TBUT in rabbits treated with or without REB-NPs. (**D**) Images of the ocular surface over the range of 0-30 min after eyelid opening in rabbits treated with or without REB-NPs. The bar indicates 1 mm. In Figure 5C,D, REB-NPs measurements were begun 6 h after the application of the formulation to the eyelid. *n* = 6. **P* < 0.05, vs. REB-MPs for each group. The application of REB-NPs induced an increase in lacrimal fluid volume, mucin levels, and TBUT in the rabbit eye and led to the stabilization of the ocular surface.

**Figure 6 pharmaceutics-12-00155-f006:**
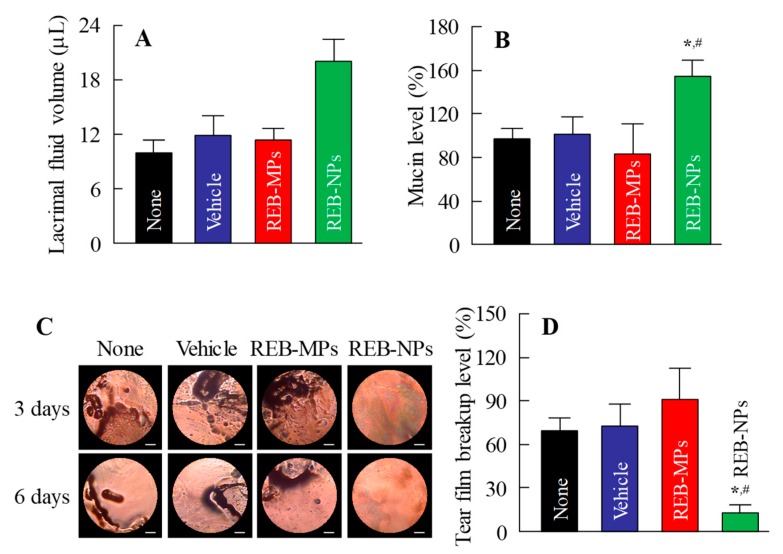
Therapeutic effect of the repetitive application of REB-NPs on dry eye in the *N*-acetylcysteine-treated rabbit model (dry eye model). Effect of REB-NPs on (**A**) the lacrimal fluid volume and (**B**) mucin levels in the dry eye model. (**C**) Images of the ocular surface in the dry eye model after repetitive applications of REB-NPs. The bar indicates 1 mm. (**D**) Effect of REB-NPs on tear film breakup levels in the dry eye model. Rabbits were treated repetitively with REB formulations at 14:00, and the experiments were performed at 18:00. *n* = 5–8. **P* < 0.05, vs. none for each group. ^#^*P* < 0.05, vs. Vehicle for each group. In the dry eye model, the application of REB-NPs enhanced the lacrimal fluid volume, and normalized the decreased mucin levels. In addition, the tear film breakup levels decreased by the application of REB-NPs.

**Figure 7 pharmaceutics-12-00155-f007:**
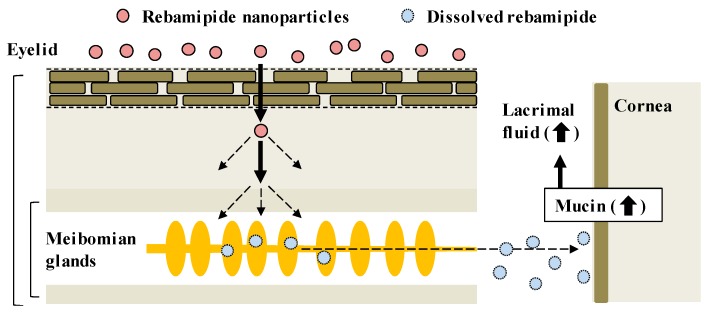
Drug delivery routes of rebamipide in REB-NPs, and the therapeutic mechanism for dry eye.

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
