# Peer review of "Novel Sustained-Release Drug Delivery System for Dry Eye Therapy by Rebamipide Nanoparticles"

_pharmaceutics, 2020, doi:10.3390/pharmaceutics12020155_

Round 1

Reviewer 1 Report

The manuscript describes nanoparticle formulation of rebamipide for dry eye treatment.

Introduction. What does this mean: ”aqueous or mucin secretagogues” ? It is not clear.

Sustained release rationale is presented in the Introduction but you should mention also how often current rebamipide formulation is applied in the clinical treatment.

Introduction. Nanoparticles are presented as a way to improve cellular delivery of drugs. This is true for some drugs that poorly permeate to the cells.  I am not sure whether this is the case for repamipide, since it is small and relatively lipophilic compounds that should enter cells easily (based on LogP and PSA).

Prevention of aggregation in the nanoparticle preparation was presented as the role of cyclodextrin in the formulation ?  Since you have cyclodextrin, some of the drug is probably in dissolved form, also forming inclusion complexes with cyclodextrin.

You should have repamipide solution control in the release test.  Otherwise, we do not know what is the real release rate. The membrane in Frantz cell causes some delay. Did you check sticking of repamipide to the membrane or other surfaces in the release test. 

Add analytical qualities of your HPLC method. 

Fig. 1D. Solubility should not be dependent on particle size, but dissolution rate depends on particle size.  Perhaps, presence of cyclodextrin causes increased solubility in the case of nanoparticles. Comment.

Fig. 3.  Did all drug dissolve in the experiment with nanoparticles ? The curve shows plateau.

Fig. 4. The unit for AUC is wrong. It should be conc x time.

Fig 5A. The lacrimal fluid volume goes up to > 20 µl in the treated group . This is strange since the normal tear fluid volume in the rabbits is 7 µl.

How was tear fluid film breakup level (%) measured ?

Rabbits have a lot of hair follicles that provide openings in the skin. Therefore, nanoparticle permeation in that case may be much higher than in the human skin. Therefore, I doubt whether this approach is clinically relevant.  Note also, that the concentrations of the drug in the tear fluid (≈ 100 µM) are only 0.1% of the initial concentration after eye drop instillation.  Thus, such concentrations might be achieved for long periods with eye drops as well. Blood flow in the eyelids removes most of the drug that permeates across the stratum corneum.  Therefore, the concentrations in the tear fluid may be so slow. Comparison with the commercial eye drop would improve this study significantly. Now comparison is only between microparticles and nanoparticles.

The authors do not have evidence to support the mechanism presented in Fig. 7. Solid particles should not permeate across the stratum corneum.  Drug dissolution might take place at some other phase of the process.

Discussion. “Meibum shows a high binding affinity to hydrophobic drugs, such as rebamipide”. This claim requires reference or some proof.

I do not believe that endocytosis would be involved in transdermal permeation of nanoparticles. Stratum corneum is a dead lipid layer without active transport processes.

Author Response

We carefully revised our manuscript according to the suggestions of the reviewer 1, and details are as follows.

< Q and A for Reviewer 1>

Q1. Introduction. What does this mean: “aqueous or mucin secretagogues” ? It is not clear.

A1. The reviewer’s comment is correct. The formulation to enhance the aqueous humor and mucin secretion were mainly used in the therapy of dry eye. In order to respond to the reviewer’s comment, we revised the sentence (line 41-42).

Q2. Sustained release rationale is presented in the Introduction but you should mention also how often current rebamipide formulation is applied in the clinical treatment.

A2. The reviewer’s comments are very important. The rebamipide were instilled 4 times/day in the clinical in the patient with dry eye. In order to respond to the reviewer’s comment, we added the information (line 51-53).

Q3. Introduction. Nanoparticles are presented as a way to improve cellular delivery of drugs. This is true for some drugs that poorly permeate to the cells. I am not sure whether this is the case for repamipide, since it is small and relatively lipophilic compounds that should enter cells easily (based on LogP and PSA).

A3. Thank you very much for pointing this out. The Biopharmaceutical Classification System (BCS) lists rebamipide as a class IV drug, and the pK and logP of rebamipide are 3.3 and 2.9, respectively. In addition, hydrophobic base was not suitable to apply the eyelid, since it is difficult to remove, and in general, the solvent to dissolve the rebamipide caused the stimulation and toxicity in the eyelid. Therefore, the nanoparticles system was usefulness to design the rebamipide formulations. In order to respond to the reviewer’s comment, we added these information in the Introduction (line 44-45).

Q4. Prevention of aggregation in the nanoparticle preparation was presented as the role of cyclodextrin in the formulation ? Since you have cyclodextrin, some of the drug is probably in dissolved form, also forming inclusion complexes with cyclodextrin.

A4. Thank you for pointing out this. The solubility of REB-MPs and REB-NPs without HPbCD were 0.003 fM, 0.011 fM, respectively. Therefore, the decrease in particle size increased the inclusion complexes with HPbCD, and it was suggested that the both of nano crystallization and enhanced inclusion complexes with HPbCD were related the increase of drug solubility in the REB-NPs. In order to respond to the reviewer’s comment, we added these data in the Results (line 177-180).

Q5. You should have repamipide solution control in the release test. Otherwise, we do not know what is the real release rate. The membrane in Frantz cell causes some delay. Did you check sticking of repamipide to the membrane or other surfaces in the release test.

A5. Thank you very much for pointing this out. Due to safety, it is difficult to prepare the carbopol gel based on dissolved rebamipide, since the solubility of rebamipide was low, and the solvent with cell toxicity need to prepare in general. In this study, we prepared the carbopol gel based on dissolved rebamipide by using the N,N-dimethylformamide as solvent. The drug release from formulation containing dissolved rebamipide was fast in comparison with REB-MPs and REB-NPs (Fig. 3S). On the other hand, in the release test, the gel formulation was applied on the membrane gently. In order to respond to the reviewer’s comment, we mention these contents in the Materials and Methods and Results, and added the data in the Fig. 3S (line 128, 207-209, Figure 3S).

Q6. Add analytical qualities of your HPLC method.

A6. The reviewer’s comment is correct. The detection limit of HPLC is 70.4 ng/ml, and the R value is 0.9992. In order to respond to the reviewer’s comment, we added the data in the Materials and Methods (line 104-105).

Q7. Fig. 1D. Solubility should not be dependent on particle size, but dissolution rate depends on particle size. Perhaps, presence of cyclodextrin causes increased solubility in the case of nanoparticles.

A7. The reviewer’s comments are very important. The solubility of REB-MPs and REB-NPs without HPbCD were 0.003 fM, 0.011 fM, respectively. Therefore, the decrease in particle size increased the inclusion complexes with HPbCD, and it was suggested that the both of nano crystallization and enhanced inclusion complexes with HPbCD were related the increase of drug solubility in the REB-NPs. In order to respond to the reviewer’s comment, we added these data in the Results (line 177-180).

Q8. Fig. 3.  Did all drug dissolve in the experiment with nanoparticles ? The curve shows plateau.

A8. Thank you for pointing out this. In the experiment using 450 nm membrane, the plateau showed that all rebamipide in the formulation sifted to reservoir side. In the reservoir side, both dissolved rabamipide and solid nanoparticles were detected, and we measured the size and number of solid nanoparticles in the Fig. 3C and D. In order to respond to the reviewer’s comment, we added the information (line 205-207).

Q9. Fig. 4. The unit for AUC is wrong. It should be conc x time.

A9. Thank you very much for pointing this out. The AUCLF for rebamipide levels in lacrimal fluid were analyzed by the trapezoidal rule up to 180 min, and we defined as “AUCLF” in the Materials and Methods. We think that it is no problem to present as “AUCLF”, since we defined in this study. Thank you for pointing out this (line 140-141).

Q10. Fig 5A. The lacrimal fluid volume goes up to > 20 µl in the treated group. This is strange since the normal tear fluid volume in the rabbits is 7 µl.

A10. The reviewer’s comment is correct. The normal lacrimal fluid volume on cornea of the rabbits is 7 µl, and 10-20 µl lacrimal fluid are observed in conjunctival sac. In this study, the Schirmer tear test strips was set to the back of the eyelid (conjunctival sac), and collected the lacrimal fluid. In order to respond to the reviewer’s comment, we added the contents in the Materials and Methods (line 138).

Q11. How was tear fluid film breakup level (%) measured ?

A11. The reviewer’s comments are very important. We mentioned the method in the Materials and Methods of pre-revised paper (line 147-151). The time from opening of the eyes to the appearance of the first dry spot in the central cornea was prevented as tear film breakup time (TBUT). The changes in tear film after winkling were monitored, and tear film breakup were evaluated by Dry eye monitor DR-1. In order to respond to the reviewer’s comment, we added the information for TBUT measurement in the Materials and Methods (line 147-151).

Q12. Rabbits have a lot of hair follicles that provide openings in the skin. Therefore, nanoparticle permeation in that case may be much higher than in the human skin. Therefore, I doubt whether this approach is clinically relevant.

A12. Thank you for pointing out this. In order to respond to the reviewer’s comment, we added the importance in the further study to target the human in the Discussion (line 321-324).

Q13. The concentrations of the drug in the tear fluid (≈ 100 µM) are only 0.1% of the initial concentration after eye drop instillation. Thus, such concentrations might be achieved for long periods with eye drops as well.

A13. Thank you very much for pointing this out. We measured the rebamipide concentration in the aqueous humor of rabbit instilled with commercially available rebamipide ophthalmic suspension (CA-REB), and the residual rebamipide was not detected 60 min after the instillation of CA-REB. In order to respond to the reviewer’s comment, we added the data in the Results (line 221-222, 308-311).

Q14. Blood flow in the eyelids removes most of the drug that permeates across the stratum corneum. Therefore, the concentrations in the tear fluid may be so slow. Comparison with the commercial eye drop would improve this study significantly. Now comparison is only between microparticles and nanoparticles.

A14. The reviewer’s comments are very important. Due to safety, it is difficult to prepare the carbopol gel based on dissolved rebamipide, since the solubility of rebamipide was low, and the solvent with cell toxicity need to prepare. In this study, we prepared the carbopol gel based on dissolved rebamipide by using the N,N-dimethylformamide as solvent, however, the application of rebamipide formulations containing dissolved rebamipide caused the redness in the eyelid of rabbit. On the other hand, in this study, we measured the rebamipide concentration in the aqueous humor of rabbit instilled with commercially available rebamipide ophthalmic suspension (CA-REB), and the residual rebamipide was not detected 60 min after the instillation of CA-REB. In order to respond to the reviewer’s comment, we added these data in the Results and Figure 3S (line 207-209, 221-222, Figure 3S).

Q15. The authors do not have evidence to support the mechanism presented in Fig. 7. Solid particles should not permeate across the stratum corneum. Drug dissolution might take place at some other phase of the process.

A15. Thank you very much for pointing this out. Our previous reports showed that the nanoparticles (< 100 nm) can penetrate the stratum corneum of skin (Ref. 13), and it is known that the thickness of eyelid skin containing stratum corneum was low in comparison with other part. In order to respond to the reviewer’s comment, we added Ref. 13 in this study. Thank you for pointing out this (Reference 13).

Q16. Discussion. “Meibum shows a high binding affinity to hydrophobic drugs, such as rebamipide”. This claim requires reference or some proof.

A16. The reviewer’s comments are very important. We showed that the rebamipide was dissolved in the meibum, since no solid nanoparticles were detected in meibum or lacrimal fluid with meibum by the dynamic light scattering measurement. These results suggested that the Meibum (lipid) showed a high binding affinity to hydrophobic drugs, such as rebamipide. In order to respond to the reviewer’s comment, we added these contents in the Results and Discussion (line 227-229, 315-316).

Q17. I do not believe that endocytosis would be involved in transdermal permeation of nanoparticles. Stratum corneum is a dead lipid layer without active transport processes.

A17. The reviewer’s comment is correct. We previously reported that the nanoparticles penetrated through the stratum corneum, and arrived to epidermis layer of skin. After that, the endocytosis related the skin penetration of nanoparticles in the epidermis layer of skin (Ref. 13). In order to respond to the reviewer’s comment, we added these contents in the Discussion (line 318-321, Reference 13).

Thank you for great comments.

Reviewer 2 Report

This manuscript aims to investigate a sustained-release drug delivery system (DDS) of rebamipide for dry eye therapy. This manuscript is well presented. However, this manuscript is missing some an important piece of evidence.  In the abstract, the authors mentioned that REB-NPs when applied to eyelids delivered rebamipide into the lacrimal fluid through the meibomian glands, and a sustained drug release was observed in comparison with CA-REB. But, no data is presented in support of this statement. The authors should do a comparative study between  REB-NPs and CA-REB to show the efficacy of REB-NP over CA-REB. In Fig.7, the authors showed that nanoparticles are crossing the eyelids. Is there a literature evidence to support this figure/representation? Why is the stability in REB formulations conducted only at 25oC? Authors could cite the articles wherein the drugs were applied to eyelids for ocular delivery and relate the study findings. It is not clear how the REB formulations were sterilized for repetitive application in the rabbit model.

Author Response

We carefully revised our manuscript according to the suggestions of the reviewer 2, and details are as follows.

< Q and A for Reviewer 2>

Q1. In the abstract, the authors mentioned that REB-NPs when applied to eyelids delivered rebamipide into the lacrimal fluid through the meibomian glands, and a sustained drug release was observed in comparison with CA-REB. But, no data is presented in support of this statement. The authors should do a comparative study between REB-NPs and CA-REB to show the efficacy of REB-NP over CA-REB.

A1. The reviewer’s comments are very important. We measured the rebamipide concentration in the aqueous humor of rabbit instilled with CA-REB, and the residual rebamipide was not detected 60 min after the instillation of CA-REB. We mentioned the result in the Discussion of pre-revised paper (line 294-297). In order to respond to the reviewer’s comment, we also added the data in the Results (line 221-222, 308-311).

Q2. In Fig.7, the authors showed that nanoparticles are crossing the eyelids. Is there a literature evidence to support this figure/representation?

A2. Thank you very much for pointing this out. In the Fig. 4C, we measured the rebamipide levels in the meibum and lacrimal fluid without meibum after the application of REB-NPs. The data showed that a little penetrated rebamipide was present in the lacrimal fluid without meibum. From these results, it was suggested that the nanoparticles are crossing the eyelids. Thank you for pointing out this (Figure 4C).

Q3. Why is the stability in REB formulations conducted only at 25oC?

A3. Thank you very much for pointing this out. In general, the ointment was stored under room temperature in the clinic. Therefore, we evaluated the dispersity and stability in REB formulations at 25℃ in this study.

Q4. It is not clear how the REB formulations were sterilized for repetitive application in the rabbit model.

A4. The reviewer’s comments are very important. The REB formulations (REB-NPs) used this study was applied to eyelid surface (not inside). Therefore, it is not need the sterilization. Thank you very much for pointing this out.

Thank you for great comments.

Round 2

Reviewer 2 Report

All comments are well addressed.